# Modulation of Synthetic Tracheal Grafts with Extracellular Matrix Coatings

**DOI:** 10.3390/bioengineering8080116

**Published:** 2021-08-20

**Authors:** Lumei Liu, Sayali Dharmadhikari, Robert A. Pouliot, Michael M. Li, Peter M. Minneci, Zhenghong Tan, Kimberly Shontz, Jed Johnson, Susan D. Reynolds, Christopher K. Breuer, Daniel J. Weiss, Tendy Chiang

**Affiliations:** 1Center of Regenerative Medicine, Abigail Wexner Research Institute, Nationwide Children’s Hospital, Columbus, OH 43215, USA; lumei.liu@nationwidechildrens.org (L.L.); Sayali.Dharmadhikari@nationwidechildrens.org (S.D.); pmminneci@gmail.com (P.M.M.); Kim.Shontz@nationwidechildrens.org (K.S.); christopher.breuer@nationwidechildrens.org (C.K.B.); 2Department of Pediatric Otolaryngology, Nationwide Children’s Hospital, Columbus, OH 43205, USA; Michael.Li@osumc.edu (M.M.L.); daniel.weiss@med.uvm.edu (D.J.W.); 3Department of Medicine, Larner College of Medicine, University of Vermont, Burlington, VT 05405, USA; Robert.Pouliot@uvm.edu; 4College of Medicine, The Ohio State University, Columbus, OH 43210, USA; ZhengHong.Tan@osumc.edu; 5Nanofiber Solutions, Inc., Columbus, OH 43017, USA; jed.johnson@nanofibersolutions.com; 6Center for Perinatal Research, Abigail Wexner Research Institute, Nationwide Children’s Hospital, Columbus, OH 43215, USA; Susan.Reynolds@nationwidechildrens.org; 7Department of Pediatric Surgery, Nationwide Children’s Hospital, Columbus, OH 43205, USA

**Keywords:** cell viability, biocompatibility, extracellular matrix coating, synthetic tracheal graft, tracheal regeneration, electrospinning, nanofibers

## Abstract

Synthetic scaffolds for the repair of long-segment tracheal defects are hindered by insufficient biocompatibility and poor graft epithelialization. In this study, we determined if extracellular matrix (ECM) coatings improved the biocompatibility and epithelialization of synthetic tracheal grafts (syn-TG). Porcine and human ECM substrates (pECM and hECM) were created through the decellularization and lyophilization of lung tissue. Four concentrations of pECM and hECM coatings on syn-TG were characterized for their effects on scaffold morphologies and on in vitro cell viability and growth. Uncoated and ECM-coated syn-TG were subsequently evaluated in vivo through the orthotopic implantation of segmental grafts or patches. These studies demonstrated that ECM coatings were not cytotoxic and, enhanced the in vitro cell viability and growth on syn-TG in a dose-dependent manner. Mass spectrometry demonstrated that fibrillin, collagen, laminin, and nephronectin were the predominant ECM components transferred onto scaffolds. The in vivo results exhibited similar robust epithelialization of uncoated and coated syn-TG patches; however, the epithelialization remained poor with either uncoated or coated scaffolds in the segmental replacement models. Overall, these findings demonstrated that ECM coatings improve the seeded cell biocompatibility of synthetic scaffolds in vitro; however, they do not improve graft epithelialization in vivo.

## 1. Introduction

The reconstruction of long-segment tracheal defects remains a challenge in airway surgery. While expansion or resection approaches are viable options for short defects of the airway, the efforts to reconstruct tracheal defects that exceed 50% in adults and 30% in children remain limited [1,2,3]. A tissue-engineered tracheal graft (TETG) that supports epithelialization could serve as a functional replacement organ. Synthetic tracheal grafts (syn-TG) have been an area of interest as they can be customized, avoid the need for donor tissue, and possess tunable mechanical properties. However, both preclinical and clinical studies using syn-TG identified two major barriers to translation: an insufficient biocompatibility and poor graft epithelialization [4,5,6].

Despite the ability to create grafts that are structurally and mechanically similar to the native trachea, syn-TG lack biochemical cues that are necessary for graft regeneration [7]. The successful repair of an epithelial defect requires the swift migration of basal epithelial progenitor cells, followed by proliferation and differentiation [8,9]. The native conditions of airway repair, the extracellular matrix (ECM), plays a critical role through the modulation of epithelial cell migration, proliferation, and differentiation [10]. Thus, ECM substrates derived from mammalian tissues are thought to mimic the in vivo tissue environment and have been used in a variety of regenerative medicine applications [11,12,13,14]. For example, ECM substrates improve the cell proliferation rates, adhesion strength, and focal adhesion on collagen- and fibronectin-mimetic surfaces in vitro [15]. Specific to the airway, ECM substrates have been used to improve the cell attachment and viability in in vitro models and to promote epithelization and prevent a chronic inflammatory reaction or secondary stenosis in an in vivo model of tracheal repair [16,17,18]. Thus, we hypothesized that scaffold modification with ECM coatings can provide critical physical and biochemical cues to promote cell viability, proliferation, and regeneration in syn-TG. To assess the performances of xenograft sources, ECM substrates derived from decellularized, lyophilized porcine and human lungs (pECM and hECM) were coated on syn-TG and characterized for their effects on the scaffold morphologies. Their effects on the cell viability and growth in vitro and a mouse model of orthotopic tracheal surgery to quantify graft epithelialization and macrophage infiltration in vivo were subsequently assessed.

## 2. Materials and Methods

### 2.1. Fabrication of a Decellularized Lung ECM Substrate

A decellularized human lung ECM substrate (hECM) was fabricated from human lung tissue obtained during an autopsy at the University of Vermont under the appropriate institutional guidelines, as previously described [14,16]. The lungs utilized were from patients with no known lung disease and either nonsmokers or remote smokers. Briefly, large cartilaginous airways were removed from whole lung sections to facilitate the complete digestion. The lung tissue was then decellularized with a continuous sequence of detergent and other solutions, including Triton-X (Sigma–Aldrich, St. Louis, MO, USA) and sodium deoxycholate (SDC, Sigma–Aldrich, St. Louis, MO, USA) [19]. The resulting acellular lung sections were frozen, lyophilized, and processed into a dry powder using a liquid nitrogen mill. This powder was weighed and resuspended in 10 mg/mL of pepsin digestion solution (1-mg/mL pepsin in 0.01-M HCl (pH = 2.00)). After the solution was digested for 72 h on a shaker table (300 rpm) using a tuning fork-style magnetic stir bar (V&P Scientific Inc., San Diego, CA, USA) at room temperature, it was centrifuged to remove the undigested and insoluble particles. The soluble ECM fraction was neutralized on ice with cold 0.1-M NaOH by thorough and gentle mixing. The resultant fraction was then lyophilized and stored at −20 °C. A decellularized porcine ECM substrate (pECM) was created following the same approach using a fresh lung obtained from a local abattoir [19].

### 2.2. Synthetic Tracheal Graft Fabrication

The synthetic tracheal grafts (syn-TG) were manufactured as previously described [4,20]. Briefly, the precursor solutions were prepared by dissolving 8 wt.% polyethylene terephthalate (PET) and 3 wt.% polyurethane (PU) in 1,1,1,3,3,3-hexafluoroisopropanol (HFIP), respectively. The PET/PU solution was then created by mixing together the solutions at 20 wt.% PET and 80 wt.% PU. The solution was electrospun on a flat sheet and cut into 2 mm × 5 mm patches for in vitro characterization. Cylindrical grafts for the in vivo tracheal replacement were created by electrospinning the PET/PU solution on a 1-mm-diameter steel rod utilizing a 20-gauge blunt tip needle. The constructs were trimmed into 5-mm grafts for long-segment tracheal replacements and cut into 1 mm × 2 mm for patch tracheoplasty. The scaffolds were sterilized by UV illumination at 35 J/cm^2^.

### 2.3. PECM and hECM Coating on Syn-TG

The PECM and hECM powders were resuspended in 0.02-M acetic acid (pH = 3.23). The resultant solution was vortexed to thoroughly dissolve the powders, yielding a 2-mg/mL stock solution. The stock solution was further diluted to 0.1-, 0.3-, 0.5-, and 0.75-mg/mL concentrations using 0.02-M acetic acid (designated as pECM 0.1, 0.3, 0.5, and 0.75 and hECM 0.1, 0.3, 0.5, and 0.75). The Syn-TG were soaked in 200-µL pECM or hECM solutions in a 48-well plate (Falcon^®^, Corning Inc., Corning, NY, USA) at 37 °C overnight on an agitator set to 200 rpm. Prior to cell seeding, the ECM coating solution was aspirated from each well and discarded. The syn-TG were rinsed twice with 500-µL 1X F-12K to remove the residual traces of acetic acid.

### 2.4. Cell Culture and Seeding

The lung epithelial cells (A549, ATCC, Manassas, VA, USA) were cultured and maintained in completed Kaighn’s Modification of Ham’s F-12 Medium (F-12K, ATCC, Manassas, VA, USA) supplemented with 10% FBS and 1% penicillin/streptomycin at 37 °C in a 5% CO_2_ incubator. To assess the effect of ECM coatings on cell proliferation in a standard cell culture, 48-well plates (0.75 cm^2^/well, Falcon^®^, Corning Inc., Corning, NY, USA) were coated with 76 µL/well of pECM or hECM using the concentrations indicated in Section 2.3. The cells (10,000 in 500-µL medium, actual volume) were seeded to either an uncoated (control) or pECM- or hECM-coated wells for 24 h. To assess the effect of ECM-coated syn-TG, 50,000 cells in 50-µL medium (actual volume) were first seeded to each of the uncoated or pECM- or hECM-coated scaffolds (2 mm × 5 mm); a higher number of cells were used to account for the increased surface area. The cells were allowed to settle for 4 h. Then, another 200 µL of culture medium was added to each well containing scaffolds, and the cells were cultured in the incubator for 24 h (24 h) or 7 days (7 d). The culture medium was exchanged every other day for 7 d.

### 2.5. Scanning Electron Microscopy

Following the incubation, uncoated or pECM- or hECM-coated scaffolds with or without seeded cells were fixed in 2.5% glutaraldehyde (Sigma–Aldrich, St. Louis, MO, USA) solution in phosphate-buffered saline (PBS, Sigma–Aldrich, St. Louis, MO, USA) for 1 h at room temperature, then rinsed three times with PBS. The samples were lyophilized for a minimum of 4 h, sputter-coated with gold-palladium, and evaluated using scanning electron microscopy (SEM, SU4800, Hitachi, Japan). The diameters of 280 individual fibers per condition were measured with ImageJ software (National Institutes of Health, Bethesda, MD, USA) from 14 different SEM images of each scaffold type. The 14 images with a clear focus on a single plane were selected from 18 images randomly captured from 6 replicates (two rounds of 3 replicates/round for each condition).

### 2.6. Cell Viability

A live/dead assay kit (Invitrogen Inc., Carlsbad, CA, USA) for mammalian cells was used to evaluate the cytotoxicity of the ECM substrate 24 h after the cells were seeded onto the control or ECM-coated tissue culture plates and to access the cell viability on the coated and uncoated syn-TG at 24 h and 7 d. The cells either in the well plates or on syn-TG were incubated in Ethidium Homodimer-1 (EthD-1, 2 µM) and Calcein-Am (1 µM) for 30 min at room temperature. EthD-1 was used to stain the dead cells red, as it can permeate cells that have lost their plasma membrane integrity. The intracellular esterase activity seen in all the live cells was stained green with Calcein-Am. The stained cells were imaged with a confocal microscope (LSM 700, Zeiss, Oberkochen, Germany). The red and green areas were measured with ImageJ software, and the live cell percentage (Live%) was calculated by (area of live cells/total area) × 100%. The total replicate number was 6 for each group, with two separate experiments (3 replicates/round).

### 2.7. DNA Assay for Cell Quantification

The number of cells present on syn-TG was quantified with a DNA extraction assay (DNeasy Blood & Tissue Kit, Qiagen, MI, USA). The cells were detached from the well plate with trypsin-EDTA, resuspended, and pipetted into capped tubes. After centrifugation at 300× *g* for 5 min, the cell pellets were resuspended in 200-µL PBS. The cells were digested by 20-µL proteinase K and 200-µL buffer AL, vortexed, and incubated at 56 °C for 10 min. The DNA was precipitated by the addition of 200-µL 96~100% ethanol and vortexed. Seeded syn-TG were incubated directly in 180-µL ATL and 20-µL proteinase K at 56 °C overnight in capped tubes. Buffer AL (200 µL) and ethanol were added, and the tubes were vortexed. 

The lysate from cultured cells or from seeded syn-TG was transferred to a DNeasy Mini spin column in a 2-mL collection tube and centrifuged at 8000 rpm for 1 min. The DNA retained in the spin column was washed with 500-µL buffers AW1 and AW2 by centrifugation at 8000 rpm (1 min) and 14,000 rpm (3 min), respectively. The DNA was eluted from the spin column by two washes of 200-µL AE buffer at 8000 rpm spinning for 1 min. 

The DNA concentration was measured using a Nanodrop^TM^ 2000c spectrophotometer (Thermo Fisher Scientific, Waltham, MA, USA). A standard curve of the cell number as a function of the DNA amount was generated using known quantities of A549 cells. The scaffolds were weighed prior to cell seeding, and the cell number per mg of scaffold was calculated. There were 9 replicates for each of the standardization groups (known cell number) and experimental groups conducted during three separate rounds of testing (3 replicates/round).

### 2.8. Mass Spectrometry

Mass spectrometry was performed on the pECM and hECM substrates immediately after lyophilization and, also, on syn-TG that were coated with pECM and hECM solutions. The samples were resuspended in 50-µL ammonium bicarbonate (ABC) solution, according to the previously described protocols [21]. For protein digestion, 5 µL of 50-mM ABC solution containing 5-µg/µL dithiothreitol was added to the sample and incubated for 15 min at 65 °C. The ABC solution containing 15-µg/µL iodoacetamide was then added, and the sample was incubated for 15 min at room temperature. Trypsin dissolved in 50-mM ABC solution was added, and the sample was digested overnight. The reaction was quenched the following day via acidification. The peptides were then dried in a vacufuge and resuspended in 20 mL of 50-mM acetic acid. 

Capillary-liquid chromatography-nanospray tandem mass spectrometry was performed using a Thermo Fisher Scientific Orbitrap Fusion mass spectrometer operated in the positive ion mode. Briefly, the samples were separated on an Easy-Spray^TM^, nano column (Pepmap^TM^ RSLC, C18, 3 µm, 100A, 75 µm × 150 mm, Thermo Fisher Scientific, Waltham, MA, USA), injected into the µ-precolumn cartridge, and desalted in 0.1% formic acid for 5 min. the samples were run at a flow rate of 300 nL/min with a spray voltage of 1.5 KV and capillary temperature of 305 °C. The entire scan was performed in Fourier-transform mode with 120,000 resolutions. The automatic gain control was set at 4 × 10^5^ ions, the maximum ion injection time was set at 50 ms, and the micro scan number was set at 1. Dynamic exclusion was enabled with a repeat count of 1 within 60 s and a low mass width and high mass width of 10 ppm.

The raw sequence information was merged using ProteoWizard and processed using Mascot Daemon, with the data searched against the Uniport Human database (version 12032015). A decoy database was used to determine the false discovery rate (FDR), and the peptides were filtered accordingly. A significance threshold was set at a *p*-value < 0.05 and proteins with a <1% FDR. The processed data was analyzed with Scaffold (Proteome Software, Inc., Portland, OR, USA), and a gene network/ontological analysis was performed using NetworkAnalyst (www.networkanalyst.ca, accessed on 25 May 2020) [22].

### 2.9. Patch Tracheoplasty and Long-Segment Tracheal Replacement

Patch tracheoplasty and a long-segment tracheal replacement with coated and uncoated syn-TG was performed on C57BL/6 mice (6~8 weeks old), as previously described [20]. Briefly, the trachea was exposed through a midline incision. A 1 mm × 2 mm anterior tracheal wall defect or a 5-mm long segment defect was reconstructed with an orthotopic implantation of same-size syn-TG substitutes. The syn-TG were coated with pECM 0.3 and hECM 0.5 mg/mL. Twelve mice were randomly assigned to coated (pECM or hECM) and uncoated patch syn-TG (*N* = 4/group) for patch tracheoplasty, and 30 mice were assigned to coated and uncoated long-segment syn-TG (*N* = 10/group). The sample sizes were determined based on our previous study and a further power analysis using an alpha at 0.5, beta at 0.2, and desired power at 0.8 [20]. The tracheal tissues were harvested for analysis at a planned experimental endpoint of 7 d. All the surgical implantation and following characterizations were performed double-blinded.

### 2.10. Histology

Tracheas bearing the implanted patch and long-segment syn-TG were explanted en bloc and immersed in 10% neutral buffered formalin solution and fixed for 48 h at room temperature. The samples were embedded in paraffin, and 4-µm axial sections were generated for patch syn-TG and longitudinal sections for long-segment syn-TG. The tissue sections were heated at 60 °C for 30 min, deparaffinized with xylene, and rehydrated with a graded ethanol series to deionized H_2_O. The sections were stained with hematoxylin and eosin (H&E, Sigma–Aldrich, St. Louis, MO, USA) using the standard methods. The stained slides were imaged with a bright field microscopy (Zeiss, Oberkochen, Germany).

Keratin 5 (K5, BioLegend, San Diego, CA, USA) was used to detect airway basal stem/progenitor cells, as previously described [8]. Briefly, the deparaffinized and rehydrated tissue sections were heated in 1× citrate buffer (Sigma–Aldrich, St. Louis, MO, USA) for 15 min for antigen retrieval. The sections were blocked in staining buffer (5% BSA and 0.1% triton X-100 in 1X PBS) at room temperature for 30 min, then incubated with the K5 primary antibody (1:1000 dilution) at 4 °C overnight. The following day, the sections were washed and incubated with a secondary antibody (donkey anti-rabbit Alexa Fluor 594, 1:500 dilution) for 45 min at room temperature. Excess antibody was removed by soaking in 1X PBS solution for 10 min and 5 dips of distilled water; then, the slides were mounted in Fluoromount-G^®^ mounting medium (Southern Biotech, Birmingham, AL, USA) for analysis and imaging. The images were obtained using fluorescent microscopy (Zeiss Imager M2, Oberkochen, Germany). A similar procedure was used to stain acetylated tubulin (ACT) for ciliated respiratory epithelial cells, with the exclusion of the antigen retrieval step. The primary antibody was mouse anti-acetylated α-tubulin (1:8000 dilution, Invitrogen), and the secondary antibody was goat anti-mouse IgG H&L (1:500 dilution, Alexa Fluor^®^ 488, Invitrogen, Waltham, MA, USA). 

The macrophages were identified by staining for CD68 using a similar approach as stated above (K5). The primary and secondary antibodies were rabbit anti-CD68 (1:200 dilution, Abcam, Cambridge, MA, USA) and goat anti-rabbit IgG H&L (1:500 dilution, Alexa Fluor^®^ 488 Invitrogen, Carlsbad, CA, USA). 

The extent of the graft epithelialization on the patch syn-TG was quantified by the measurement of the epithelium on the luminal surface. The circumferential width of the syn-TG surface was demarcated by perpendicular lines at the boundaries of the native trachea. Understanding that the epithelialization may vary along the length of the patch (see Results, Section 3.4), five regions of the patch syn-TG lumen were quantified, representing the proximal anastomosis (zone 1), proximal midgraft (zone 2), midgraft (zone 3), distal midgraft (zone 4), and distal anastomosis (zone 5). The zones were determined by sectioning through the trachea axially and evenly distributing the total slides into 5 groups. The first, middle, and last slides in each zone were stained with H&E, and the slides adjacent to the middle slides were stained with immunofluorescence. They were all imaged with a microscope, and the representative images were randomly selected and are shown in the Results section. The overall percent of graft epithelialization of each animal was calculated as an aggregate of these zones by using the following equation:$$\frac{\sum_{1}^{5}\left(width\;of\;syn-TG\;lumen\right)-\sum_{1}^{5}\left(width\;of\;non-epithelialized\;syn-TG\;lumen\right)}{\sum_{1}^{5}\left(width\;of\;syn-TG\;lumen\right)}\times 100\%$$

The basal cell (K5+) and ciliated cell (ACT+) quantification of the neo-epithelium were measured from IF images with ImageJ and represented by the percentage of the lumen covered by the neo-epithelium. The experimental width was measured by the width between the edges of wound in the experimental region. The covered width and experimental width were measured in 5 sections evenly distributed between the first and last sections of the defect. The K5 and ACT coverage percentages along the luminal surface of the patch (% coverage) were calculated as in a similar manner as stated above:$$\frac{\sum_{1}^{5}\left(covered\;width\right)}{\sum_{1}^{5}\left(experimental\;width\right)}\times 100\%$$

The epithelialization of long-segment syn-TG was measured on the longitudinal plane. This was performed by first identifying the slides that represented the true sagittal midpoint of the host trachea and graft [20]. The percent of the syn-TG lumen with neo-epithelium was compared and quantified with both H&E and immunofluorescence. 

The macrophages (CD68+) in the epithelial submucosa of patch and long-segment syn-TG were calculated as the positive cells per high power field (HPF) area of 100 µm × 100 µm when compared with a nonprimary antibody (negative) control on the adjacent slide [20].

### 2.11. Statistical Analyses

The data were analyzed with GraphPad Prism 8 (GraphPad Software, San Diego, CA, USA). The PET/PU fiber diameters were compared using the nonparametric Kruskal–Wallis test. The cell viability and seeded cell numbers were expressed as the mean ± standard deviation (SD). The normality of the data was evaluated using the Shapiro–Wilk test. The data were compared with multi-one-way ANOVA between the control (uncoated) and pECM- or hECM-coated TETG. The median with a 95% confident interval (CI) was used in the fiber diameter measurements. A linear correlation between the coating concentrations and cell number/mg synthetic scaffold was analyzed using GraphPad Prism 8. The statistical significance was considered as *p* < 0.05 (two-tailed).

## 3. Results

### 3.1. Decellularized ECM Substrates Alter the Fiber Thickness of Electrospun Scaffolds and Surface Topography

The effect of ECM coating on the electrospun nanofiber scaffold morphology was first assessed. Coating syn-TG resulted in the preservation of the porosity of the electrospun nanofiber scaffold (Figure 1A,C). A scaffold coating with pECM or hECM did not alter the surface topography of the fibers at lower concentrations (0.1~0.5 mg/mL); however, higher concentration coatings (0.75 mg/mL) resulted in visible hydrogel formation on the nanofibers (yellow arrows). ECM-coated fibers exhibited higher fiber diameters despite the absence of visible debris (all concentrations except for pECM 0.1 mg/mL) (Figure 1B,D).

### 3.2. PECM and hECM Coating Improves Seeded Cell Viability and Proliferation on Syn-TG In Vitro

The effect of ECM coating on the cell viability in vitro (Figure 2) was then assessed. In standard cell culture conditions in well plates, the viability of seeded cells on the uncoated and pECM- and hECM-coated wells were uniformly high (Figure 2A(a)). At the 24-h and 7-d time points, pECM- and hECM-coated syn-TG were qualitatively found to have a greater proportion of living cells with increased cell density on the syn-TG scaffolds (Figure 2B). Quantitatively, pECM (0.3) and hECM (0.5 and 0.75) improved the viability at 24 h (Figure 2C). PECM (0.3) and hECM (0.5) were also found to enhance the cell proliferation by 24 h (Figure 2A(b)).

At 7 days, all the coating concentrations improved the cell viability on syn-TG in vitro (Figure 2C). The viability did not improve with higher concentrations; pECM (0.75) revealed a decrease from 24 h to 7 d. Similar to our findings at 24 h, pECM (0.3 and 0.75) and hECM (0.5 and 0.75) were both found to improve the cell proliferation at 7 days (Figure 2D). Unlike its effects on the viability, the effect of coating on cell proliferation appears to be dose-dependent and improves at higher concentrations (Figure 2D). The trend lines indicate the positive correlation of both pECM and hECM coatings with the cell number at 24 h and 7 d. 

### 3.3. PECM and hECM Are Predominantly Composed of Collagen, Fibrillin, and Laminin

Next, we then characterized our matrix substrates with mass spectrometry. Overall, there were 25 and 18 protein families identified in pECM and hECM, respectively. However, of these proteins, only six families from pECM and six families from hECM were identified on the coated syn-TG scaffolds (Table 1), suggesting an incomplete adherence of the coating to the TG scaffolds. Among the coated peptides, collagen, fibrillin, laminin, and nephronectin were found on both the pECM- and hECM-coated scaffolds. Notable peptides that were found in the coating but not detected on the scaffolds include: glycosaminoglycans, fibrinogen, and elastin. In contrast, keratin was not found in the coating but was detected on the coated fibers and was determined to be a contamination. It was excluded from the analysis.

The functional pathways of the scaffold peptides were analyzed with a protein interaction analysis via NetworkAnalyst (Table 1). In both pECM (a) and hECM (b), the predominant cellular pathways were found to be related to cell migration and proliferation, namely: ECM–receptor interactions, focal adhesion, protein digestion, and the PI3K-Akt signaling pathway (Table 1).

### 3.4. Syn-TG Patches Exhibit Graft Epithelialization and Macrophage Infiltration

The effect of ECM coating on syn-TG epithelialization was then assessed in vivo. The coating concentrations for the tracheal patches were selected based on the concentrations that best improved the viability and cell number in vitro (pECM 0.3 mg/mL and hECM 0.5 mg/mL). Orthotopic implantation of the syn-TG patches did not result in respiratory distress, and the survival rate was 100% (12 out of 12). Histologically, all the groups preserved the graft patency without evidence of stenosis, fibrosis, or collapse. Cellular coverage was observed on the graft in all groups by 1 week (Figure 3). The center of the patch had less epithelium than the edge (Figure 3B). The basal and ciliated cell quantifications exhibited similar patterns of coverage. There was no difference in the overall coverage of the patch with the epithelial cells between uncoated and pECM and hECM (75.3% ± 5.5%, 67.3% ± 21.8%, and 80.4% ± 2.6%, respectively, Figure 3D). The epithelialization—specifically, of the basal cells (K5+) and multi-ciliated cells (ACT+)—was substantial at 1 week in all the groups (coverage with K5+ epithelium: uncoated—90.1% ± 17.2%, pECM 0.3—89.0% ± 9.9%, and hECM 0.5—91.1% ± 10.8%; coverage with ACT+ epithelium: uncoated—60.9% ± 15.2%, pECM 0.3—40.9% ± 20.3%, and hECM 0.5—42.5% ± 25.2%) (Figure 4A). 

Syn-TG patch implantation resulted in macrophage infiltration (Figure 4B). The macrophages were primarily identified in the submucosa of the neo-epithelium. The ECM coating did not have an effect on the macrophage infiltration.

### 3.5. Long Syn-TG Exhibit Poor Graft Epithelialization and Higher Macrophage Infiltration Correlating with Higher Coating Concentration

With a favorable graft epithelialization in the patch tracheoplasty model, the performance of the ECM-coated scaffolds in a mouse model of a segmental tracheal replacement was then assessed using pECM-coated scaffolds. Two higher concentrations of pECM (0.5 and 0.75 mg/mL) were selected for coating these grafts in order for us to assess the effect of the substrate in the presence or absence of a hydrogel formation (Figure 5A). The animals implanted with uncoated, pECM 0.5-, and pECM 0.75-coated syn-TG all exhibited a 60% survival at 7 d (six out of 10). All the groups preserved the graft patency without evidence of a collapse, however, in contrast to the patch tracheoplasty model, syn-TG exhibited poor overall epithelialization irrespective of the ECM coating (coverage with ACT+ epithelium: uncoated—4.9% ± 5.6%, pECM 0.5—9.9% ± 8.6%, and pECM 0.75—13.8% ± 11.8%) (Figure 5 and Figure 6A,B). Given that epithelialization was limited in syn-TG, the macrophage infiltration was quantified in the host epithelial submucosa adjacent to the graft (Figure 6C). pECM 0.75 syn-TG exhibited higher macrophage infiltration than pECM 0.5 (*p* = 0.0348).

## 4. Discussion

Tissue-engineered trachea have the potential to replace absent or diseased tissues in the management of long-segmented tracheal defects. However, the preclinical and clinical experiences with TETG have highlighted the barriers to its translation, including chronic inflammation, delayed epithelialization, and graft stenosis [23]. While synthetic scaffolds are able to recapitulate the mechanical properties of the native trachea and provide physical support for cell regeneration, they lack the biochemical cues that can reside in the native ECM [7]. ECM hydrogels have been used to model the ECM in vitro, improve tissue-engineered constructs, and are clinically applied in vivo [14]. We hypothesized that the coating of syn-TG with an airway-specific ECM would improve the cell viability, growth, and graft epithelialization while attenuating the macrophage infiltration. To study the effect of the ECM on a synthetic scaffold, we used coating materials derived from the lungs rather than the trachea, as the trachea proved impossible to fully decellularize and digest into a soluble powder. 

The ECM coating was found to increase the scaffold fiber diameter while preserving the surface topology; visible hydrogel formations were observed at the highest concentrations tested. Fiber diameter and porosity have been found to influence the cell proliferation in vitro; we did not explore this contribution on the PET/PU fibers [24,25,26,27]. Our coated scaffolds exhibited improved cell viability on syn-TG in vitro in stark contrast to the poor viability seen in the uncoated scaffolds. In addition to the improved cell viability, a dose-dependent increase in the recovered cell count from syn-TG suggests that an ECM coating enhances the cell proliferation on syn-TG [28].

Using mass spectrometry, we found that a substantial proportion of the hECM and pECM proteins were successfully coated on the synthetic scaffolds—most notably, collagen and laminin (Table 1). Collagen and laminin are critical extracellular structures for cell migration during early repair, can simulate the production of TGF-β for the modulation of cell adhesion, and support basement membrane synthesis [10,29,30,31,32]. Other dominant proteins in ECM coating include fibrillin and nephronectin. Fibrillin is a major component of microfibrils, and the elastic fibers found in connective tissue and nephronectin are an ECM protein involved in cellular adhesion and repair [33,34,35]. The proteins that failed to coat the scaffold (notably glycosaminoglycans and elastin) may have been rinsed off during the initial coating procedure, cell culture, and processing for mass spectrometry due to their highly hydrophilic properties [36]. In addition, elastin is readily degraded by detergents and is often not found in decellularized constructs [19]. Glycosaminoglycans and elastin play critical roles in cell adhesion, proliferation, and wound repair [37,38]. The future work will include the effects of the successful coating of these proteins on the performance of syn-TG.

Informed by the in vitro findings, we then assessed the effect of ECM coating with a mouse model of orthotopic tracheal graft implantation by using models of syn-TG patches [20]. We implanted segmental syn-TG coated with higher concentrations of pECM (0.5 and 0.75 mg/mL) to access the effect of the substrate in the presence or absence of a hydrogel formation. Unlike our in vitro findings, the coated syn-TG scaffolds did not enhance the graft epithelialization (Figure 4 and Figure 5). This disparity between the in vitro and in vivo findings may be due to several factors. First, the improved viability and cell growth of the lung epithelium following the scaffold modulation with the lung ECM in vitro may not have the same effect on the tracheal epithelium. Second, the A549 cell line is derived from adenocarcinomic human alveolar basal epithelial cells, which may have different affinities to the coated scaffold compared to tracheal epithelial cells. There are limitations in deriving matrix substrates from large airways, thus leading to our use of lung-derived coatings. We speculate that the lower epithelialization on the segmental syn-TG compared to the patch syn-TG is likely due to (1) the surface area in the segmental replacement being larger than the area seen in our patch model and (2) having only two epithelial fronts (proximal and distal) on the long segmental grafts only allowed longitudinal cell migration as opposed to circumferential fronts on the patches, which allowed both circumferential and longitudinal cell migration. Our current long-segmented syn-TG were not able to support the epithelialization, as well as neovascularization in vivo. The revascularization of a trachea substitute is believed to be a critical step towards a successful tracheal transplantation [39,40,41]. Our future work will be devoted to assessing the capacity for revascularization in tracheal grafts.

As a secondary outcome measure, we hypothesized that the introduction of the ECM coatings would attenuate the macrophage infiltration. Our previous work demonstrated an increased macrophage infiltration with syn-TG implantation [20]. Macrophages drive the host response to biomaterials, such as innate immune responses and acute rejection [42,43,44,45,46]. In our study, similar levels of macrophage infiltration were observed in the submucosa over the pECM0.3- and hECM0.5-coated and uncoated patches (Figure 5). This finding may be a function of the timing of the explant (7 d) or may suggest that the ECM coatings did not enhance or attenuate the immune response in the syn-TG patch. In the larger scale defect model, the syn-TG coated with pECM 0.75 mg/mL showed higher macrophage infiltration (Figure 6). This finding is supported by studies that found that a higher concentration of ECM hydrogel coating introduced acute macrophage infiltration in the brain tissue in a rat model of a stroke [47,48]. ECM hydrogel can also introduce a macrophage phenotype shift from the M1 to the M2-like anti-inflammatory phenotype thus facilitating tissue repair [49,50]. Future studies will include more robust attempts to create tracheal hydrogels and the investigation of macrophage phenotypes responding to different concentrations of lung ECM coatings.

Successful airway repair with synthetic tracheal grafts has yet to be achieved. Synthetic tracheal grafts have not demonstrated the capability to accommodate the surgical, mechanical, and infectious/inflammatory considerations seen in tracheal replacements [41]. However, novel approaches to composite tracheal grafts, combining biologic and synthetic components, may introduce a new role for synthetic materials in tracheal replacements.

## 5. Conclusions

Lung-derived ECM coatings are nontoxic and improve the epithelial cell attachment on syn-TG scaffolds in a dose-dependent manner in vitro. We believe that critical ECM proteins are conserved between species; specifically, collagen and laminin improve the cell proliferation rates and focal adhesion on the syn-TG surface. In vivo, ECM coatings are innocuous and nonimmunogenic for patch syn-TG, and ECM hydrogel induced more macrophage infiltration into long-segmented syn-TG. Additional studies are needed to define the benefits of ECM hydrogels on syn-TG for tracheal regeneration.

## Figures and Tables

**Figure 1 bioengineering-08-00116-f001:**
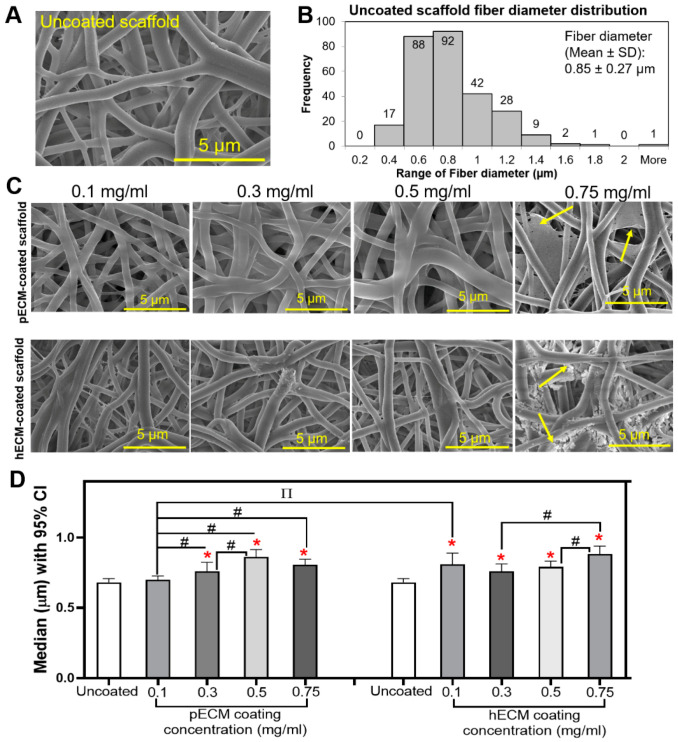
Characterization of the tracheal scaffolds. (**A**) SEM image of syn-TG. (**B**) Fiber diameter distribution (frequency) of uncoated syn-TG (distributions of the coated scaffolds are not shown). (**C**) SEM images of scaffolds coated by 0.1-, 0.3-, 0.5-, and 0.75-mg/mL pECM and hECM. Arrows denote hydrogel formation in pECM and hECM 0.75-mg/mL-coated fibers. (**D**) Median diameter (µm) of the fibers with a 95% confidence interval (CI). * Represents the differences of all the coating groups, except pECM 0.1 mg/mL, compared with the uncoated scaffold (*p* = 0.0011 for pECM (0.3); *p* = 0.0064 for hECM (0.3); *p* < 0.0001 for pECM (0.5 and 0.75); and hECM (0.1, 0.5, and 0.75)). # Represents the differences between different concentrations within the same coating type (*p* = 0.0477 for pECM (0.1) vs. (0.3), *p* < 0.0001 for pECM (0.1) vs. (0.5), *p* < 0.0001 for pECM (0.1) vs. (0.75), *p* = 0.0052 for pECM (0.3) vs. (0.5), *p* = 0.0001 for hECM (0.3) vs. (0.5), and *p* = 0.0041 for hECM (0.5) vs. (0.75)). ∏ Represents the differences between the different coating types, pECM and hECM, within the 0.1-mg/mL concentration (*p* = 0.0003).

**Figure 2 bioengineering-08-00116-f002:**
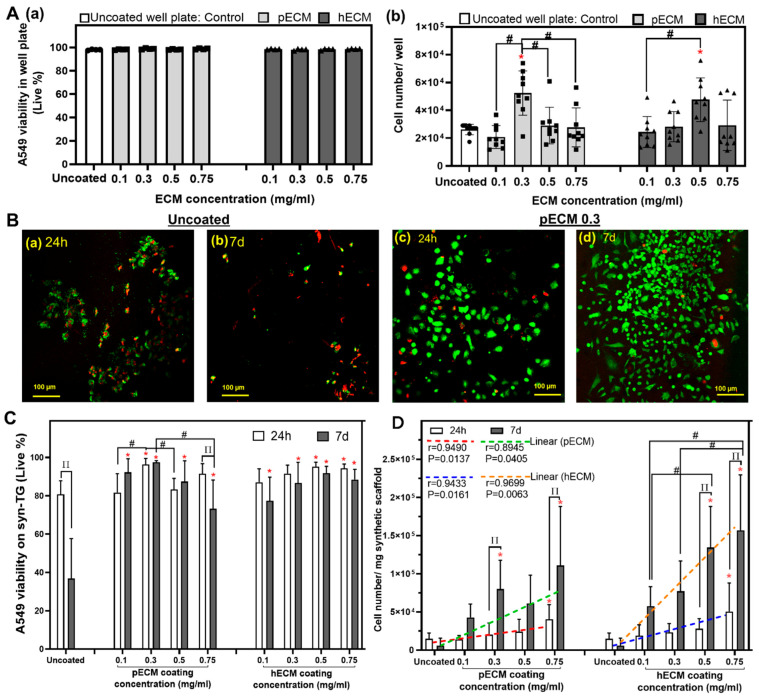
Cell viability and cell numbers in the pECM- and hECM-coated well plates and syn-TG in vitro. (**A**) The impact of pECM and hECM coatings on the cell viability and recovery in 2D well plates. (a) Viability of the cells in uncoated and coated well plates. No effect was seen at 24 h. (b) Recovered cell numbers in the 2D well plate. * Represents that the coated well plate with pECM 0.3 mg/mL and hECM 0.5 mg/mL increased the proliferation compared with the uncoated well plate (*p* = 0.0019 and 0.0223, respectively). # Represents a cell number increase in the pECM 0.3-mg/mL-coated well plate compared to the 0.1-, 0.5-, and 0.75-mg/mL-coated well plates (*p* < 0.001 and *p* = 0.0084 and 0.0044, respectively). # Represents the increase in cell number in the hECM 0.5-coated well plate compared to the hECM 0.1-coated well plate (*p* = 0.0092). (**B**) Representative images of live and dead cells on uncoated ((a),(b)) and pECM 0.3-mg/mL-coated scaffolds ((c),(d)) at 24 h and 7 d (other live/dead images can be found in Appendix A in the Appendix A). (**C**) Viability quantification of the cells on the uncoated and coated scaffolds at 24 h and 7 d. ∏ A decline in the viability was noted in the uncoated and pECM 0.75 mg/mL from 24 h to 7 days (*p* < 0.0001 and *p* = 0.494, respectively). # The increased viability of pECM 0.3 compared to pECM 0.1 and pECM 0.5 at 24 h (*p* = 0.0019 and 0.0081, respectively); at 7 d, there was a lower cell viability with the pECM 0.75-coated scaffold compared with the pECM 0.3-coated scaffold (*p* = 0.0162). * Represents the improvements in the viability of the coated scaffolds at 24 h compared to the uncoated (pECM 0.3, *p* < 0.001; hECM 0.5, *p* = 0.0026; and hECM 0.75, *p* = 0.0057); at 7 d, all the concentrations of the pECM and hECM coatings enhanced the cell viability compared to the uncoated scaffold (*p* < 0.001). (**D**) The cell numbers on the scaffold after 24 h and 7 d. At 24 h, * the scaffolds coated with pECM 0.75 and hECM 0.75 mg/mL showed higher cell numbers compared with the uncoated (*p* = 0.0387 and 0.0059, respectively). At 7 d, the scaffolds coated with pECM 0.3 and pECM 0.75 and hECM 0.5 and hECM 0.75 mg/mL all demonstrated higher recovered cell counts compared with the uncoated scaffold (*p* = 0.0282 and *p* < 0.001 for pECM and *p* < 0.0001 for hECM). There was a correlation between the concentration of both the pECM and hECM coatings with the cell number at 24 h (pECM: r = 0.9490, *p* = 0.0137; hECM: r = 0.9433, *p* < 0.0161) and at 7 d (pECM: r = 0.8945, *p* < 0.0405; hECM: r = 0.9699, *p* < 0.0063).

**Figure 3 bioengineering-08-00116-f003:**
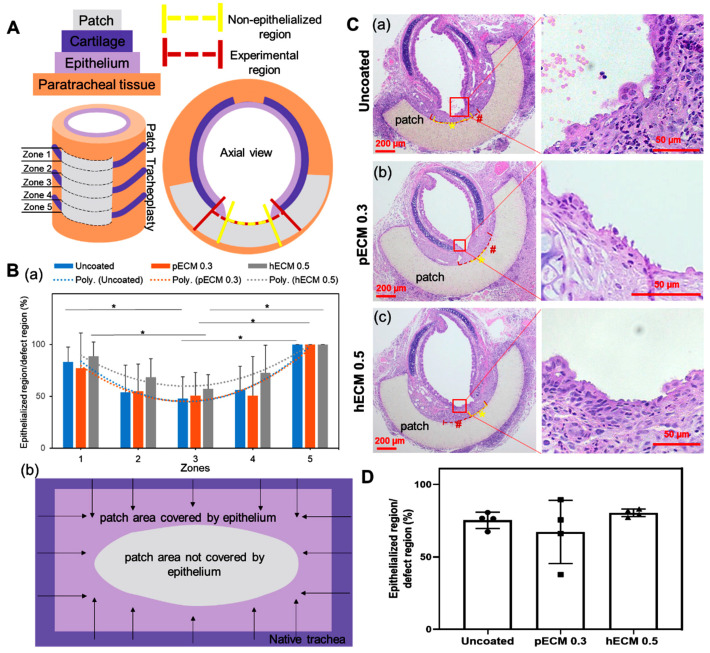
The epithelialization of the pECM- and hECM-coated patches of syn-TG in vivo. (**A**) Schematic definition of the regions for the epithelium coverage calculations. (**B**) Regional cell coverage. (a) Quantification of the overall cell coverage in 5 regions and the second degree of the polynomial regression curves. * Denotes the differences between zone 3 and zone 1 and zone 3 and zone 5 (uncoated zone 1 vs. zone 3: *p* = 0.0366, uncoated zone 3 vs. zone 5: *p* = 0.0155, pECM 0.3 zone 3 vs. zone 5: *p* = 0.0217, hECM 0.5 zone 1 vs. zone 3: *p* = 0.0179, and hECM zone 3 vs. zone 5: *p* = 0.0082). (b) A schematic top view of the patch luminal surface indicated by the quantification; arrows denote the epithelial regeneration of the patch. (**C**) Representative cross-section H&E images of a trachea section with a patch implanted: (a) uncoated, (b) pECM 0.3-mg/mL, and (c) hECM 0.5-mg/mL-coated patches. * Represents the noncovered lengths of the dashed lines along the patches. # Represents the experimental (defect) lengths of the longer dashed lines along the patches. Regenerated cell length was calculated by experimental (defect) length minus non-covered length. (**D**). Quantification of overall cell coverage calculated by the regenerated cell width/experimental width × 100%.

**Figure 4 bioengineering-08-00116-f004:**
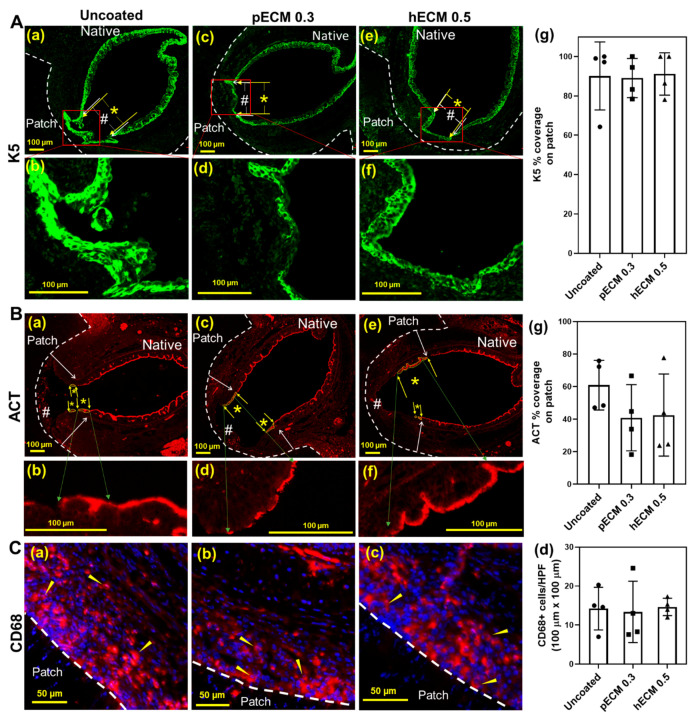
Immunofluorescence characterization of a basal epithelial cell (K5), ciliated epithelial cell (ACT), and macrophage (CD68) on patch syn-TG. (**A**) Representative IF images of basal epithelial cells (K5, green, (a)~(f)) in tracheas implanted with uncoated, pECM 0.3-mg/mL, and hECM 0.5-mg/mL-coated patches (scale bar = 100 µm); dashed lines mark the edges of the patches; in (a),(c),(e), * denotes the width of the neo-epithelium; # denotes the total width of the intraluminal syn-TG patch between the arrows. (g) Quantified coverage of the patch with basal (K5) (% coverage). (**B**) Representative IF images of ciliated epithelial cells (ACT, red, (a)~(f)) in (a),(c),(e); * and # are identical nomenclature for the coverage with ciliated epithelium (ACT). (g) Quantified coverage of the patch with ciliated (ACT) cells (% coverage). (**C**) Representative IF images of CD68+ macrophages (arrowheads mark the red cells) in the submucosa over the patch: uncoated (a), a pECM 0.3-mg/mL-coated patch (b), and a hECM 0.5-mg/mL-coated patch (c). (d) Quantification of the macrophages (cells/100 µm × 100 µm area) in the neo-epithelium.

**Figure 5 bioengineering-08-00116-f005:**
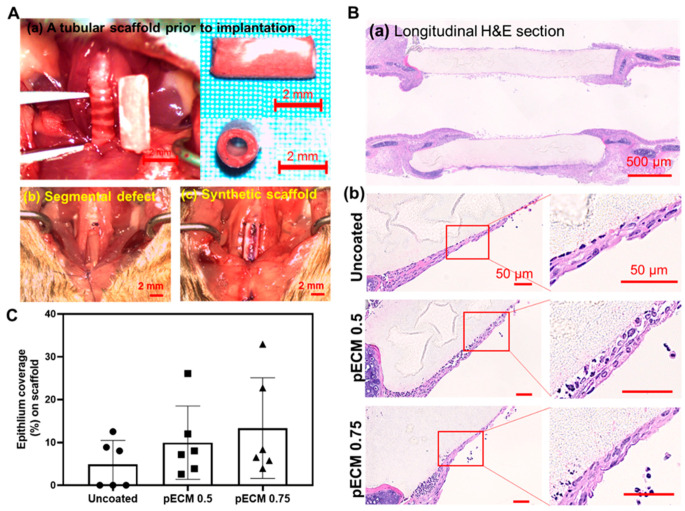
Long-segment synthetic tracheal graft implantation. (**A**) Atubular syn-TG was trimmed to 5 mm prior to the implantation (a), and then, a 5-mm segmental defect (b) was reconstructed with syn-TG (c). (**B**) Representative H&E images of the (a) longitudinal section of the trachea with a syn-TG, and (b) the luminal surface of the implanted uncoated and pECM 0.5-mg/mL and pECM 0.75-mg/mL-coated syn-TG at 7 d. (**C**) Quantification of the overall epithelial coverage on syn-TG.

**Figure 6 bioengineering-08-00116-f006:**
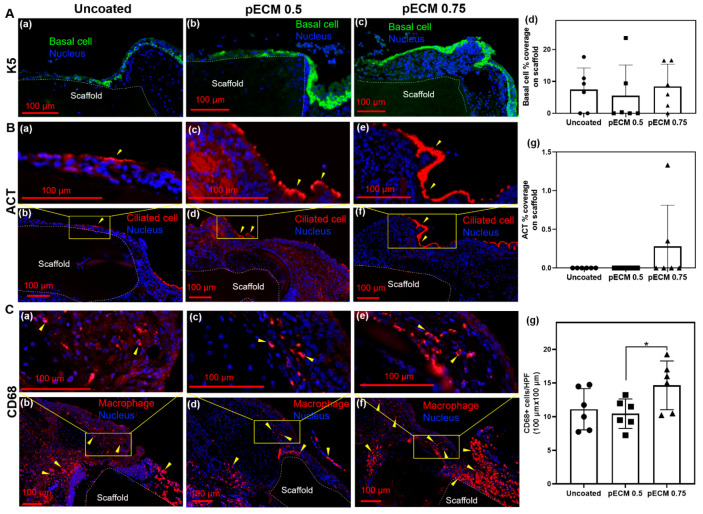
Immunofluorescence characterization of the basal epithelial cell (K5), ciliated epithelial cell (ACT), and macrophage (CD68) on long-segmented syn-TG. (**A**) Representative images (a)~(c) and the quantification (d) of the K5 coverage on the lumen of uncoated and pECM 0.5- or 0.75-mg/mL-coated syn-TG. (**B**) Representative images (a)~(f) and quantification (g) of the ACT coverage on the lumen of uncoated and pECM 0.5- or 0.75-mg/mL-coated syn-TG. (**C**) Representative images of the macrophage infiltration in the tracheal submucosa surrounding syn-TG (a)~(f), and the quantification of the macrophage infiltration (g). * Represents pECM 0.75-mg/mL-coated scaffolds that exhibited higher macrophage infiltrations than pECM 0.5 (*p* = 0.0348). No difference was detected between the pECM 0.75-coated scaffold and the uncoated control.

**Table 1 bioengineering-08-00116-t001:** The proteins and functional pathways identified by the protein interaction analyses in pECM and pECM-coated syn-TG and hECM and hECM-coated syn-TG.

(a) pECM and pECM-coated syn-TG						
Identified Proteins in pECM and pECM-Coated PET/PU		Gene ID	ECM–Receptor Interaction	Focal Adhesion	Protein Digestion/Absorption	PI3K-Akt Signaling Pathway
Fibrillin 1&2		FBLN1&2				
Acidic mammalian chitinase isoform c		CHIA				
C-type lectin domain-containing protein		CLEC				
Nephronectin		NPTN				
Laminin (LAM)	Laminin subunit alpha-5	LAMA5	√	√		√
	Laminin subunit beta-1	LAMB1	√	√		√
	Laminin subunit alpha-3	LAMB3	√	√		√
	Laminin subunit gamma-1	LAMC1	√	√		√
	Laminin subunit gamma-2	LAMC2	√	√		√
Collagen (COL)	Collagen alpha-1(I) chain	COL1A1	√	√	√	√
	Collagen alpha-2(I) chain	COL1A2	√	√	√	√
	Collagen alpha-1(II) chain	COL2A1	√	√	√	√
	Fibrillar collagen	COL11A1&2			√	
	Collagen alpha-2(IV) chain	COL4A2	√	√	√	√
	Collagen alpha-3(VI) chain	COL4A3	√	√	√	√
	Collagen alpha-1(V) chain	COL5A1			√	
**(b) hECM and hECM-coated syn-TG**						
**Identified Proteins in hECM and hECM-Coated PET/PU**		**Gene ID**	**ECM–Receptor Interaction**	**Focal Adhesion**	**Protein Digestion/Absorption**	**PI3K-Akt Signaling Pathway**
Fibrillin-1		FBN1				
Metalloproteinase inhibitor 3		TIMP3				
Antileukoproteinase		SLPI				
Nephronectin		NPNT				
Laminin (LAM)	Laminin subunit alpha-3	LAMA3	√	√		√
	Laminin subunit alpha-5	LAMA5	√	√		
	Laminin subunit beta-1	LAMB1	√	√		
	Laminin subunit beta-2	LAMB2	√	√		
	Laminin subunit beta-3	LAMB3	√	√		
	Laminin subunit gamma-1	LAMC1	√	√		
Collagen (COL)	Collagen alpha-1(I) chain	COL1A1	√	√	√	√
	Collagen alpha-1(III) chain	COL3A1			√	
	Collagen alpha-1(IV) chain	COL4A1	√	√	√	√
	Collagen alpha-1(V) chain	COL5A1			√	
	Collagen alpha-2(I) chain	COL1A2	√	√	√	√
	Collagen alpha-2(IV) chain	COL4A2	√	√	√	√
	Collagen alpha-2(V) chain	COL5A2			√	
	Collagen alpha-3(VI) chain	COL6A3	√	√	√	√

## Data Availability

The data presented in this study are available on request from the corresponding author. The data are not publicly available due to privacy.

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
