# Peer review of "Modulation of Synthetic Tracheal Grafts with Extracellular Matrix Coatings"

_bioengineering, 2021, doi:10.3390/bioengineering8080116_

Round 1

Reviewer 1 Report

Liu et al describe how porcine ECM and human ECM affects the biocompatibility and epithelialization of synthetic tracheal grafts (syn-TG) both in vitro and in vivo. To address the two translational gaps of syn-TG: insufficient biocompatibility and poor graft epithelialization, the authors hypothesized that scaffold modification with ECM coatings can provide critical biochemical and physical cues to promote cell viability, proliferation, and regeneration in syn-TG. By fabricating pECM and hECM coated syn-TGs and experimental methods such as scanning electron microscopy, cell seeding, cell viability testing, DNA assay, mass spectropy, and histology examination, the authors reported the following findings: ECM coatings increased the scaffold fiber diameter compared to synthetic fibers at higher concentrations. ECM coatings were not cytotoxic and rather enhanced in vitro cell viability and growth on syn-TG in a dose-dependent manner. They implanted coated or uncoated patches or 5 mm segments in mice,  and explanted after 7 days. The coating did not improve epithelialization in the in vivo segmental replacement models.

Major comments:

  • In Result Section 1.1, the authors stated that decellularized ECM substrates preserved porosity of the scaffold fibers. However, from Figure 1C, it could be observed that some hydrogels created by hECM and pECM at 0.75 mg/mL blocked some of the pores in the fiber structure. This could impact the porosity of the scaffold. To address this issue, the authors should set up tests to quantify the porosity of scaffolds and notify the reader on whether there is a difference.
  • In Figure 1D, the y-axis is labelled with median. However, median was not the data that was mentioned in the methods to quantify fibre diameters. It was mean. Thus, the author should either change the description in the method section to “median” or make another bar graph that demonstrate mean as bar height and standard deviation as error bar. Authors should also explain why the trend line is placed there and the meaning of the parameters (slope and y-intercepts) of the trendline.
  • Figure 2B. Authors should show the images for other pECM concentrations, as well as hECM. At the lease these should be included as part of supplementary data.
  • Quantification of proteins in the ECM and coating might be useful to compare the ECM with native lung and native trachea. Especially since in vivo epithelialization with tracheal epithelium was poor compared to in vitro (possibly due to the use of lung ECM?).
  • The authors also pointed out that the adherence of certain critical components of ECM (fibrinogen and elastin) to the synthetic grafts was not great. What are the implications to in vivo results? Could the lack of these critical components result in the poor epithelialization data in vivo? Perhaps the authors can simply coat with these factors rather than the whole ECM. Or if they truly believe that the entire ECM is needed for synergistic effects, then focus should be on ensuring adherence of these key components.
  • It is unclear why the authors chose the higher porcine ECM concentrations (0.5 and 0.75 mg/mL) for all of the in vivo segment implantations? The patches were coated with 0.3 mg/ml pECM or 0.5 mg/ml hECM as per the performance of the respective concentrations in vitro.
  • It is unclear why pECM was chosen over hECM in the log segment studies. Is the functionality of the ECM dependent on species? One possibility is to test ECM derived from mice. These may perhaps perform better in vivo? The authors mention the use of allografts in the study. However all of the models presented are xeno.
  • Authors analyzed 5 zones on the patches, but also indicated that within the middle zones (2-4) the epithelialization varied (being more epithelialized toward the edges, poor epithelialization in the centre). A better description of where the pictures were taken from is required. It is also unclear how many images were taken per section. It seems that 2 rounds were done with 3 replicates per round but a total of 14 images were taken from all of these? Why 14? And how were the 14 selected?

Minor comments:

  • In Section 2.4 there is a typo. “… wells were coated with 76 μl/well of pECM or hECM using concentrations indicated in section 4 2.3.”

  • In Section 2.4, the author said 10,000 in 500 μl medium were seeded to wells and 50,000 cells in 50 μl medium were seeded to scaffolds. The author should clarify whether these numbers are concentrations or actual volume of cell-containing medium used.

  • In Section 1.4, the sentence ­­“There was no difference in overall coverage of the patch with epithelial cells between uncoated, pECM and hECM (75.3 ± 5.5 %, 67.3 ± 21.8% and 80.4 ± 2.6 %, respectively)” should reference Figure 3D to keep the consistency of the format.

  • Other minor comments:

Location

Comment

Line 41

Remove comma after functional

Line 62

“coated on” instead of “coated to”

Line 206

Add “and” after last comma.

Line 254

“high power field area (HPF)” because they use the HPF abbreviation in later figures.

Figure 1 caption

p values not given for * and # (unless the (p<0.001) at the end refers to *, #, and ?)

Line 293-294

They are specifically referring to pECM 0.75 mg/ml but they broadly say “pECM”.

Line 310 (Fig 2)

“uncoated and pECM” not “at”

Line 314 (Fig 2)

“compared to uncoated”

Figure 4 Title

Should be bold to match the other figure titles.

All figures

Inconsistent graph symbols:

Figure 2: circles for both uncoated and pECM, squares for hECM.

Figure 3: circles for uncoated, squares pECM, triangles hECM.

Figure 4A(g): circles for all (uncoated, pECM, hECM).

Figure 4B(g) and C(g), 5, 6: circles for uncoated, squares pECM, triangles hECM.

Figure 5: circles for uncoated, squares pECM, triangles hECM.

Figure 6 Title

Should be bold to match the other figure titles.

Line 412 (Fig 6)

 (a)-(f), not (a)-(c)

Line 413 (Fig 6)

(g) instead of (d).

Line 452-453

Missing part of a sentence? “Instead, we observed that syn-TG 452 patches rea This disparity between…”

Line 475

Figure 6.

Reviewer 2 Report

See attached comments in PDF

Reviewer 3 Report

The subject of the present research is sound, since at the moment there is an imperious need to improve the biocompatibility of the synthetic tracheal implants/ grafts. The text of the manuscript is comprehensive and sections are appropriately developed. The methodology is accurately described. The manuscript should be accepted for publication after minor revision.

  1. Lines 110-113: it is not clear which is the deposited cell number/ surface area.
  2. 'Cell attachment' (Lines 283, 294-297) and cell adhesion (Line 435) are mentioned. However, the cells survival or cells no. after days of incubation does not represent the cell attachment or adhesion, since this is rather related to an interphase and refers to the type as well as the extent of the biochemical, mechanical or electrostatic connection between cells and substrates [Kevin V. Christ & Kevin T. Turner (2010) Methods to Measure the Strength of Cell Adhesion to Substrates, Journal of Adhesion Science and Technology, 24:13-14, 2027-2058, DOI: 10.1163/016942410X507911]. Authors may refer to cells proliferation instead or simply cells no.
  3. Typing error - Line 453.
  4. Conclusions are poor and could be improved so that they summarize better the interesting findings. 

Round 2

Reviewer 1 Report

Authors have adequately addressed the comments/queries